# Absence of the RING domain in *MID1* results in patterning defects in the developing human brain

Sarah Frank[1],*, Elisa Gabassi[1],*, Stephan Käseberg[2] , Marco Bertin[2], Lea Zografidou[2], Daniela Pfeiffer[2], Heiko Brennenstuhl[3] , Sven Falk[1],† , Marisa Karow[1],† , Susann Schweiger[2],†

The X-linked form of Opitz BBB/G syndrome (OS) is a monogenic disorder in which symptoms are established early during embryonic development. OS is caused by pathogenic variants in the X-linked gene *MID1*. Disease-associated variants are distributed across the entire gene locus, except for the N-terminal really interesting new gene (RING) domain that encompasses the E3 ubiquitin ligase activity. By using genome-edited human induced pluripotent stem cell lines, we here show that absence of isoforms containing the RING domain of MID1 causes severe patterning defects in human brain organoids. We observed a prominent neurogenic deficit with a reduction in neural tissue and a concomitant increase in choroid plexus-like structures. Transcriptome analyses revealed a deregulation of patterning pathways very early on, even preceding neural induction. Notably, the observed phenotypes starkly contrast with those observed in MID1 full-knockout organoids, indicating the presence of a distinct mechanism that underlies the patterning defects. The severity and early onset of these phenotypes could potentially account for the absence of patients carrying pathogenic variants in exon 1 of the *MID1* gene coding for the N-terminal RING domain.

## Introduction

With the discovery that mutations in the *MID1* gene located in Xp22 are the underlying reasons for the development of Opitz BBB/G syndrome (OS) (OMIM 300000), a causative genetic link has been established decades ago (Quaderi et al, 1997). OS is a rare congenital malformation syndrome characterized by a multitude of symptoms including developmental delay and brain anomalies such as hypoplasia of the cerebellar vermis and the corpus callosum (Baldini et al, 2020). Biochemical dissection of the MID1 protein structure uncovered three zinc-binding domains with the E3 polyubiquitin ligase activity provided by the RING (really interesting new gene) domain, whereas the BBox1 and BBox2 domains were shown to convey auto-monoubiquitination E3 ligase activity (Han et al, 2011; Kaur et al, 2023). Besides *MID1*, disruption of several other proteins containing E3 ubiquitin ligase activity has been associated with neurodevelopmental disorders (Upadhyay et al, 2017; Ebstein et al, 2021), for example, *HUWE1* (Froyen et al, 2008) and *UBE3A* (Kishino et al, 1997), underscoring their significance for brain development. One of the targets of the MID1 E3 ubiquitin ligase activity is the catalytic subunit α4 of protein phosphatase 2A, thereby directly linking MID1 with the mTORC1 signaling pathway (Trockenbacher et al, 2001; Liu et al, 2011). Further studies connected the MID1/PP2A/mTORC1 axis with the subcellular localization of GLI3, a mediator of the sonic hedgehog signaling (SHH) pathway, demonstrating the importance of MID1 in regulating the SHH pathway (Baldini et al, 2020). Indeed, MID1 targets FU for proteasomal degradation; hence, absence of FU results in reduced expression of SHH pathway target genes by lowering nuclear translocation of GLI3 (Schweiger et al, 2014). Besides the E3 ligase activity, MID1 has been shown to interact with microtubules (Schweiger et al, 1999), dependent on its phosphorylation status (Liu et al, 2001). The impact of a complete absence of the multifaceted MID1 protein or deletion of specific domains of MID1 on human neurodevelopment has remained largely understudied, primarily because of the lack of appropriate model systems to adequately address this question. Until recently, studies using human model systems focusing on the neural phenotypes observed in patients have been limited to two-dimensional cultures. With the advent of using human induced pluripotent stem cells (hiPSC) to generate brain organoids, we can now shed light on the molecular mechanisms underlying neurodevelopmental disorder phenotypes within a more intricate cellular environment, modeling

---

[1]Institute of Biochemistry, Friedrich-Alexander-Universität Erlangen-Nürnberg, Erlangen, Germany   [2]Institute of Human Genetics, University Medical Center of the Johannes Gutenberg University Mainz, Mainz, Germany   [3]Institute of Human Genetics, Heidelberg University, Heidelberg, Germany

Correspondence: marisa.karow@fau.de
Elisa Gabassi's present address is Department of Genomics, Stem Cell Biology and Regenerative Medicine, Institute of Molecular Biology and CMBI, Leopold-Franzens-University Innsbruck, Innsbruck, Austria
*Sarah Frank and Elisa Gabassi contributed equally to this work
†Sven Falk, Marisa Karow, and Susann Schweiger contributed equally to this work and co-corresponding authors

early human brain development (Velasco et al, 2020; Kelley & Pasca, 2022).

## Results and Discussion

High clinical variability between OS patients has been described repeatedly (summarized in Winter et al [2016]), but interestingly, when taking a closer look at the documented pathogenic variants within the *MID1* gene, no homogenous distribution of mutations along the gene body is found (Fig 1A). In fact, no pathogenic variants in the sequence encoding the E3 ubiquitin ligase activity harboring the RING domain have been reported so far. Computational predictions of the REVEL (rare exome variant ensemble learner) score (Ioannidis et al, 2016) as readout for the putative pathogenicity of single nucleotide variants revealed high pathogenicity scores localizing to the RING domain (Fig 1A) indicating that genetic variation in this segment of the gene has a higher probability of causing deleterious effects on protein function (Schroter et al, 2023). To functionally test the impact of mutations in the region of the *MID1* gene with a high predicted pathogenicity score and, at the same time, complete absence of reported patients, we used male hiPSCs and engineered hemizygous mutations in the coding exon 1 of the *MID1* gene. This CRISPR/Cas9-mediated approach gave rise to two distinct genome-edited *MID1* RING domain variant hiPSC lines, referred to as Rm1 and Rm2 (RING domain mutated 1 and 2). As depicted in Fig 1B, in Rm1, we introduced a 2-base pair (bp) deletion c.204_205delAG (Fig S1A), whereas in the Rm2 hiPSC line, genome editing resulted in a 1-bp insertion c.136_137insC (Figs 1B and S1A). Both modifications did not result in a change of *MID1* mRNA abundance, neither C-terminally nor N-terminally, when compared with isogenic control cells, showing that the introduced changes in the *MID1* gene did not trigger nonsense-mediated mRNA decay (Fig 1C). However, in both variants, a premature stop codon was generated because of the introduced frame-shift, resulting in a loss of the 75 kD full-length MID1 protein produced from ATG1 (Fig 1D, arrow). In controls and in Rm1 and Rm2 additional polypeptides translated from ATG3-5 giving rise to 64, 58, and 57 kD MID1 proteins, respectively, were detectable, whereas a 69 kD isoform is exclusively produced in Rm1 mutant cells because of the 2-bp deletion that puts the ATG2 in-frame with the rest of the protein (Fig 1B and D). Common to both Rm hiPSC lines is the absence of the full-length MID1 protein. Importantly, the RING finger domain is present only in the full-length MID1 protein and absent in all N-terminally truncated variants translated from ATG2-5, whereas ATG4 and ATG5 in addition lack the B-Box1 domain (Fig 1B and D). To determine the subcellular localization of these MID1 variants, we generated different GFP-tagged *MID1* fusion constructs (Fig 1F) and expressed those in HeLa cells. In addition to the N-terminally truncated *MID1* variants, we also engineered a GFP fusion of a patient-specific *MID1* variant harboring a C-terminal 4-bp deletion (Fig 1E) in the coding exon 9 referred to as del4 (Schweiger et al, 1999). In concordance with the role of the B30.2 domain (encoded by exons 8 and 9) for microtubule association (Schweiger et al, 1999), the del4 variant showed no filamentous organization and no co-localization with the microtubule cytoskeleton indicating that the del4 variant lost

its microtubule association. In contrast, all the other variants expressed from ATG1-5 exhibited unaltered filamentous configuration patterns and co-localized with the microtubule cytoskeleton indicating maintained association with microtubules (Fig 1F).

Next, we asked how mutations affecting the RING domain might impact early human brain development and generated brain organoids using an unguided organoid formation protocol (Lancaster et al, 2013). We also included an isogenic male genome-edited hiPSC line exhibiting a full knock-out of the complete *MID1* gene, hereafter referred to as KO hiPSC line (Fig S1B). First morphological phenotyping of the embryoid bodies (EB) 3 d after aggregation of hiPSCs revealed a significant reduction in EB size in the *MID1* RING domain variants Rm1 and Rm2 compared with the EBs generated from their isogenic control (Ctrl) hiPSCs and opposite to the EBs generated from the KO hiPSC line which were bigger in size (Fig S1C and D). This size difference was not maintained until d30, as no apparent differences in the overall size could be detected anymore (Fig 1G). Inspection of the tissue organization in d30 organoids revealed striking differences with Rm1 and Rm2 organoids showing fewer ventricular zone-like structures (VZLS) containing neural stem and progenitor cells as compared with the Ctrl organoids (Fig 1G and H). Brain organoids generated with the *MID1* KO hiPSC line showed the opposite phenotype with an increase in VZLS in the KO organoids (Fig 1G and H), showing that the phenotypes observed in Rm1 and Rm2 are not caused by a loss of MID1 function but suggest a different underlying mechanism. To assess whether increased cell death could be the cause of a diminished organization into VZLS, we stained for the apoptosis marker activated Caspase 3 (actCAS3) (Velasco et al, 2019) but did not detect any apparent differences between the d30 organoids generated from the different experimental hiPSC lines (Fig S1E). Next, we localized the expression of MID1 protein in organoid slices, using an antibody targeting the N-terminal portion of the MID1 protein, that is, amino acids 84–113 were used as immunogen. As highlighted in the scheme (Fig 1E), this antibody recognizes the full-length MID1, expressed only in Ctrl cells, and polypeptides derived from ATG2, only expressed in Rm1, but not the shorter isoforms produced from ATG3-5. Congruently, we obtained a positive signal in organoid slices derived from the Ctrl, and the Rm1 hiPSC lines, but not from the Rm2 hiPSCs. Moreover, no signal was detected in *MID1* KO organoids (Fig 1I), confirming the specificity of the antibody. We then used this antibody against the N-terminus to address the dynamics of MID1 protein expression from young (d30) stages with many well discernible neural stem cell, containing VZLS to older organoids (2 mo) with more developed and mature neuronal compartments (Fig S1F). Together, these experiments not only show MID1 expression in neural stem and progenitor cells residing in the VZLS during early human neural development, thereby extending earlier findings that showed Mid1 expression in mouse neural stem and progenitor cells (Dal Zotto et al, 1998; Pinson et al, 2004) but also reveal MID1 protein expression in postmitotic neurons.

We next assessed the impact of the Rm mutations on tissue composition. To this end, we stained d30 organoids for the neural stem cell marker SOX2 and the neuronal marker MAP2. The area in an organoid slice covered by either SOX2 or MAP2 was used as a measure to quantify how much neural tissue is present in a brain organoid. By assessing the relative fraction of neural tissue per

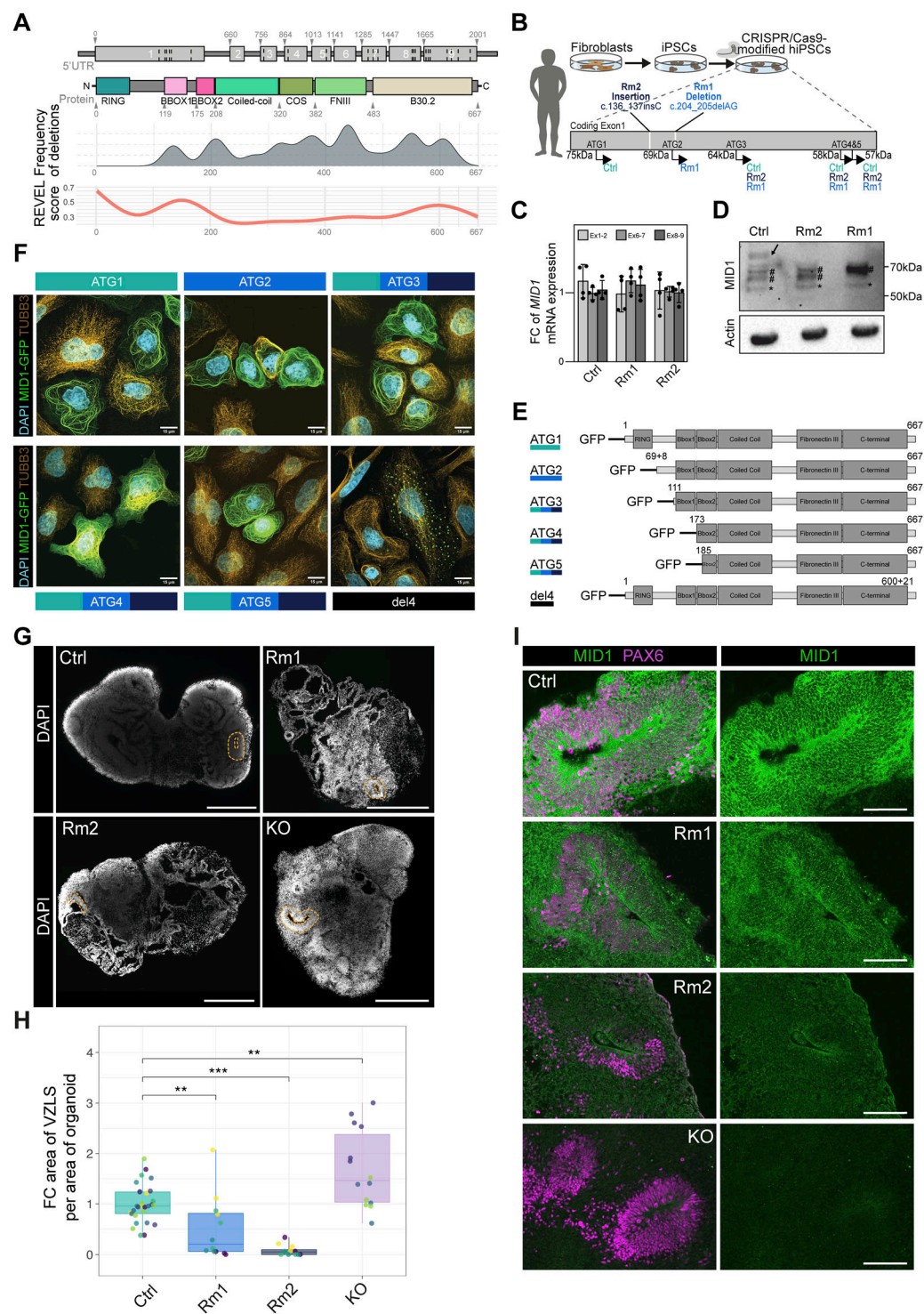

**Figure 1. Generation of mutant *MID1* human induced pluripotent stem cell (hiPSC) lines and organoids.**
**(A)** Schematic depicting the *MID1* gene including exons. Numbers indicate base pairs. Dashed lines indicate reported mutations in the *MID1* gene (Winter et al, 2016). Protein structure including major domains and amino acids. Below the MID1 protein structure, the frequency of deletions along the gene is summarized. The lowest panel indicates the rare exome variant ensemble learner (REVEL) score along the *MID1* gene body. Note the peak of the REVEL score in the N-terminal end of MID1, correlating with the absence of reported pathogenic variants in patients. **(B)** Scheme highlighting the experimental procedure to derive fibroblasts from a male healthy donor and perform CRISPR/Cas9-mediated perturbations in coding exon 1 of *MID1* in the hiPSCs derived from these fibroblasts. The resulting Rm1 and Rm2 mutations are caused by a 1-bp insertion and a 2-bp deletion, respectively. Besides the full-length MID1 protein that is produced when translation starts from ATG1, further alternative ATGs and their expected usage in the different MID1 hiPSC lines are schematized. **(C)** Quantitative RT–PCR of the expression levels of *MID1* using specific primers to detect exons 1–2, 6–7, and 8–9 across Ctrl, Rm1, and Rm2 hiPSC lines (n = 4). **(D)** Western blot showing MID1 protein expression using an antibody against the C-terminus of MID1. Actin is

total organoid, we detected a marked reduction in the neural fraction in Rm but not KO organoids compared with Ctrl organoids (Fig 2A and B), suggesting differences in the efficacy of neural induction during organoid formation. When we then further dissected the composition of the neural tissue, we detected a reduction in the neuronal area covered by MAP2 relative to the total neural area (MAP2 and SOX2) in Rm but not in KO organoids (Fig 2A and C). Hence, specifically in the Rm lines, earlier neural induction processes are defective, and later neuronal differentiation deficits occur (Fig 1G and H). In line with the observation on VZLS, these data show contrasting phenotypes between the full knockout of MID1 and Rm mutants, in which only the full-length but not the shorter isoforms are lost.

MID1 is expressed very early during embryonic development (Dal Zotto et al, 1998; Pinson et al, 2004; Baldini et al, 2020), and given the difference already in the EB size observed on d3 after aggregation of the hiPSCs, we set out to examine molecular changes induced by the different *MID1* mutations starting at the hiPSC state. We performed bulk RNA-sequencing (RNA-seq) of Ctrl and *MID1* variant lines at hiPSC stages and 5, 8, and 11 d after induction of the organoid differentiation protocol (Fig S2A). As shown in a heatmap depicting the normalized counts per million (CPM) values, pluripotency genes such as *OCT4* (*POU5F1*), *SOX2*, *NANOG*, and *SALL4* were unaffected in the different hiPSC lines. Similarly, also the expression of *MID1* and the *MID1* homologue *MID2* were unaffected in the Rm mutant lines, confirming earlier qRT–PCR data (Fig 1C), whereas *MID1* was not detected in KO cells (Fig S2B). We next analyzed the impact of the Rm mutations on the expression dynamics of *MID1* early during neural differentiation, showing the above-described early induction of *MID1* expression independent of whether the lines carry Rm mutations or not (Fig S2C). Beyond these genes, the data revealed a noticeable difference in the global transcriptome between the experimental lines already at the hiPSC stage. As evident from the principal component analysis (PCA) and highlighted through 95% confidence ellipses, the respective biological triplicates of the Rm1 and Rm2 hiPSC lines cluster together and are diverging from both the Ctrl and KO clusters (Fig 2D). Together, these findings offer compelling evidence that both the neural tissue phenotypes and the changes in the molecular landscape early on are akin between the two Rm lines but quite distinct between Rm and KO organoids.

To uncover the molecular machinery responsible for the observed early neural induction deficits (Fig 2B), we dissected the transcriptomic consequences induced by the Rm mutations in *MID1* from the hiPSC state throughout neural induction and early differentiation into brain organoids (d5, d8, d11). PCA revealed that PC1 separates the samples according to their developmental stage along neural differentiation, whereas PC2 describes divergence between the experimental conditions, yet again with the different Rm lines clustering together with high confidence (Fig 2E). Next, we determined genes significantly up- (Fig 2F) or downregulated (Fig 2G) after departure from hiPSC state towards early differentiation in each line (Table S1) and compared the resulting gene sets between the lines to find similarities and discrepancies in Ctrl and Rm mutants. This revealed not only failure to induce the expression of genes associated with proper neural induction in the Rm lines, that is, 162 genes upregulated in Ctrl samples but not in Rm samples (Fig 2F in bold), but also lack of downregulation of genes possibly impairing neural differentiation in Rm lines, that is, 78 genes downregulated in Ctrl samples but not in Rm samples (Fig 2G in bold). We furthermore assessed the genes which were aberrantly induced in the Rm variant samples, that is, 97 genes shared by Rm1 and Rm2 (Fig 2F in bold), and those genes that were aberrantly lost in the Rm samples, that is, 87 genes (Fig 2G in bold), and performed gene ontology (GO) analyses. Whereas GO terms indicating proper neural induction such as "dendrite development," "neuron projection morphogenesis," or "telencephalon development" were induced in the Ctrl samples (Fig 2H), GO terms such as "anterior/posterior pattern specification," "embryonic morphogenesis," or "tissue morphogenesis" were significantly enriched when we analyzed the aberrantly induced genes in the Rm lines (Fig 2I). The GO terms of selectively downregulated genes during neural induction in the Ctrls, that is, 78 (Fig 2G), or jointly in the Rm lines (87 as indicated in Fig 2G), are summarized in Fig S2D and E, respectively. These indications of an early patterning phenotype prompted us to further address the (mis)expression of patterning genes. Indeed, we found a marked upregulation of *HOXD3*, *HOXB4*, and *HOXB5* (Figs 3A and S2F), members of the HOX gene family (Luo et al, 2019; Saito & Suzuki, 2020) in the Rm lines, indicating a caudalization in the *MID1* Rm variant organoids. Furthermore, expression of *GBX2* (gastrulation brain homeobox 2) (Fig 3A), a gene involved in the establishment of the midbrain/hindbrain boundary and expressed in the posterior part of the embryo early on (Wassarman et al, 1997), indicates patterning towards more caudal brain regions. These data are supported by a later induction of *PTF1A* (pancreas-associated transcription factor 1A), highly expressed in hindbrain neuronal

used as a loading control. The arrow indicates full-length MID1, whereas # indicates truncated MID1 proteins of 69, 64, and 58/57 kD, and the star indicates an unspecific band. **(E)** Scheme depicting full-length and N-terminally truncated MID1 proteins resulting from the usage of alternative ATGs. The lowest scheme depicts the gene structure of a patient-derived *MID1* variant exhibiting a 4-bp deletion at the C-terminal end. The color code on the left indicates lines in which this isoform is present (turquois: Ctrl, blue: Rm1, purple: Rm2, black: del4). **(E, F)** Micrographs showing cellular co-localization of MID1 isoforms fused to GFP with TUBB3 (orange) after overexpression of *MID1-GFP* constructs (summarized in (E)) in HeLa cells. The color of the box above (upper panel) or below (lower panel) the pictures indicates presence of this isoform in the respective hiPSC lines (turquois: Ctrl, blue: Rm1, purple: Rm2, black: del4). Note the formation of aggregates and loss of microtubule association upon overexpression of MID1 with a 4-bp deletion in the C-terminus as shown previously (Schweiger et al, 1999). Scale bar = 15 µm. **(G)** Images of representative Ctrl, Rm1, and Rm2 d30 brain organoid slices show the cellular organization through DAPI staining. The yellow dashed lines highlight representative ventricular zone-like structures (VZLS). **(H)** Quantification of the areas of VZLS covering the total area of brain organoid slices shown as box plots with jitters indicating individual d30 organoids. The data reveal a decrease in the contribution of VZLS to the brain organoids in the MID1 Rm organoids. Dots represent individual organoids derived from different batches, as indicated by distinct colors. Ctrl: n = 29 from seven batches, Rm1: n = 12 from four batches, Rm2: n = 12 from four batches, KO: n = 14 from three to six batches. Mann–Whitney–*U* test. **P < 0.01, ***P < 0.001. Exact *P*-values (top to down) 0.0017, 6.5 × 10$^{-7}$, 0.0027. Boxplots show median, quartiles (box), and range (whiskers). **(I)** Immunofluorescence stainings show the expression of MID1 (green) and PAX6 (magenta) in d30 brain organoid slices. MID1 signal can be detected in Ctrl and Rm1, but not in Rm2 or KO brain organoids. For (F, G), DAPI was used to counterstain nuclei. For (G, I), scale bar = 100 µm.

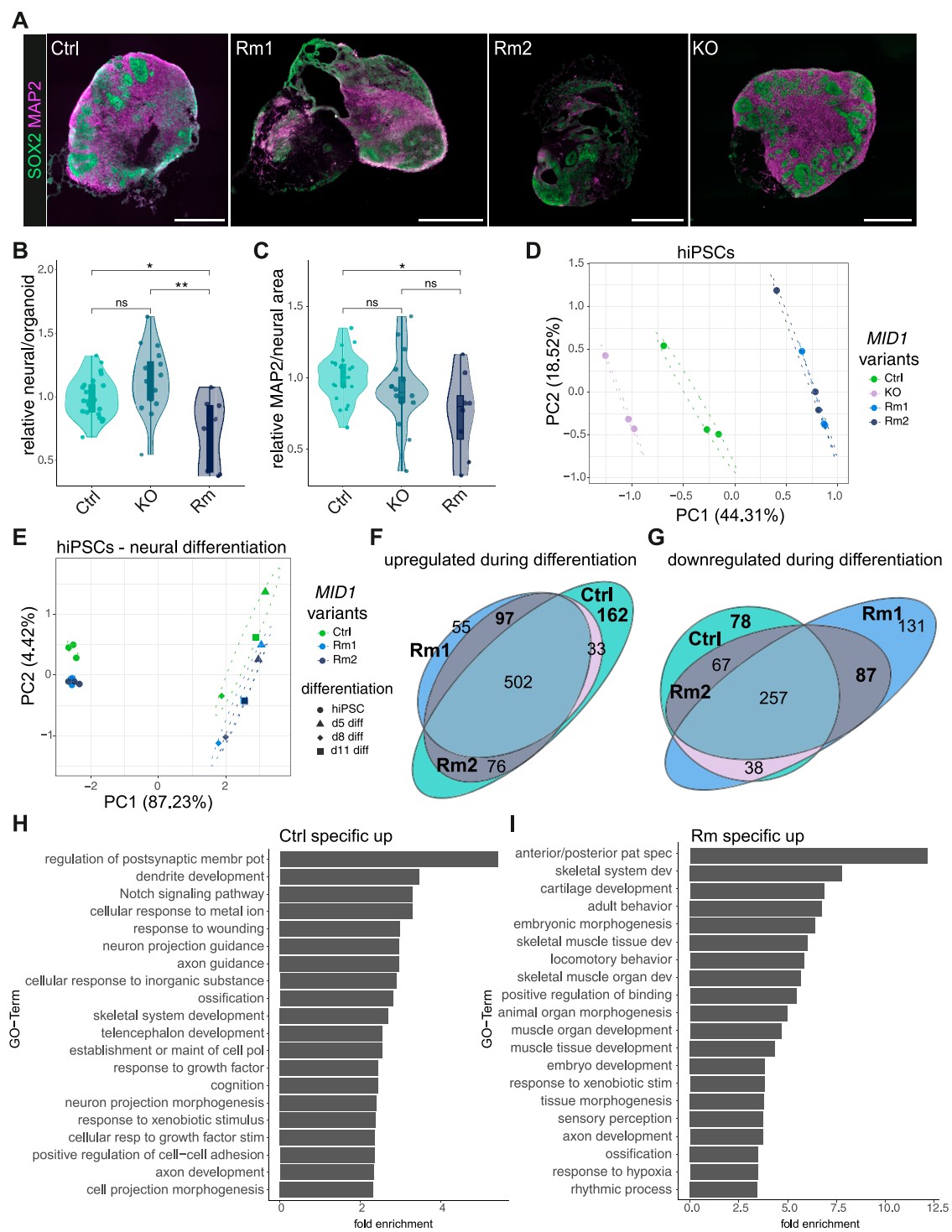

**Figure 2. Cellular and molecular characterization of brain organoids derived from *MID1* variant human induced pluripotent stem cells (hiPSCs).**
**(A)** Immunofluorescence stainings of brain organoid slices showing SOX2 (green) and MAP2 (magenta) positive cells in d30 Ctrl, Rm1, Rm2, and KO organoids. DAPI was used to counterstain nuclei. Scale bar = 500 $\mu$m. **(B)** Quantification of the relative contribution of neural areas as quantified by the fraction of SOX2+ and MAP2+ per total area in d30 organoids reveals a decrease in the MID1 Rm organoids compared with Ctrl and KO organoids (Rm = Rm1 + Rm2), as shown by violin ad jitter plots. Mann-Whitney-$U$ Test. Exact $P$-values (top to down) 0.01, 0.0017, 0.057. **(C)** Within the neural area, quantification of MAP2 in d30 organoids revealed reduced neural differentiation in the MID1 Rm organoids, as shown by violin and jitter plots. Mann-Whitney-$U$ test. Exact $P$-values (top to down) 0.02, 0.11, 0.15. For (B, C), *$P < 0.05$, **$P < 0.01$, ns, not significant. Dots represent individual organoids. Ctrl: n = 26 from six independent batches, Rm: n = 14 from six independent batches, KO: n = 16 from three independent batches. **(D)** Principal component analysis segregated the transcriptomes of the experimental hiPSC as highlighted by 95% confidence ellipses (dashed lines). **(E)** The PC plot depicts the transcriptional divergences of the Ctrl, Rm1, and Rm2 hiPSC lines from the hiPSC state throughout early differentiation into brain organoids (d5, d8, d11 of differentiation). Confidence ellipses (dashed lines) illustrate that Rm lines cluster together but differ from Ctrls. **(F)** Euler diagram showing the

cells (Hoshino et al, 2005) (Fig 3A). In addition, we also found increased expression of *CHRD* (chordin), expressed in more dorsal structures (Bachiller et al, 2000) (Fig 3A), and a concomitant incapacity to induce the expression of markers for anterior and ventral CNS domains such as *DLX5* (Distal-less Homeobox 5) (Liu et al, 1997; Yang et al, 1998) (Fig S2G). To dissect potential signaling pathways involved, we focused our attention on SHH signaling, given that SHH signaling is not only a crucial player in early CNS patterning but also MID1 has been shown to regulate SHH signaling via ubiquitination of FU (Schweiger et al, 2014). In line with these previous observations, we found that at early differentiation stages, the Rm lines failed to upregulate *GLI1* (GLI Family Zinc Finger 1), *PTCH2* (Patched 1), and *SHH* (Fig 3B), known modulators of the SHH pathway. Furthermore, *SCUBE2* (signal peptide, CUB domain, and EGF-like domain containing 2), a gene repressed by SHH signaling in the early neural tube (Collins et al, 2023 Preprint), is higher expressed in the differentiating Rm lines as compared with Ctrl (Fig 3B). To extend these analyses beyond SHH signaling and survey the impact of the Rm mutations on general signaling pathways involved in cell-to-cell communication in a temporally resolved manner, we computed a normalized transcriptional deviation score for specific KEGG signaling pathways (Kanehisa et al, 2023) in Rm mutants to their respective control at each analysis timepoint. This analysis revealed a strikingly early divergence of the retinol metabolism implicated in retinoic acid signaling and the TGF-β signaling pathway before the changes observed in the SHH pathway (Fig 3C). Both of these pathways have been implicated in the early induction of ventral fates independent of SHH (Patten et al, 2003; Placzek & Briscoe, 2005; Meinhardt et al, 2014), suggesting that the primary effect on patterning in the *MID1* Rm mutants is not exclusive to changes in SHH signaling but includes early alterations in retinoic acid and TGF-β signaling.

To correlate the early deregulation of signaling pathways with the later regionalization during brain organoid development, expression levels of genes indicating regional identity were determined in d30 organoids by qRT-PCR. We detected in d30 Rm organoids a misexpression of patterning genes, such as an upregulation of markers of the dorsal most region, the choroid plexus, such as *BMP6* (Bone Morphogenetic Protein 6), *LMX1A* (LIM Homeobox Transcription Factor 1 Alpha), *OTX2* (Orthodenticle Homeobox 2), and *TTR* (Transthyretin) (Fig 4A). None of these transcripts were upregulated in the organoids derived from the *MID1* KO hiPSC line, again highlighting different mechanisms between the *MID1* KO and Rm organoids. Concomitant with the upregulation of dorsal markers, we found a decrease in the ventrally expressed patterning gene *PTCH1* (Fig 4A), indicating decreased ventral-sourced SHH signaling (Murone et al, 1999) in Rm organoids also at this later stage of organoid differentiation (d30) (see also Fig 3B). To correlate the higher expression of dorsal

markers with the reduced expression of ventral markers, we performed paired qRT-PCR on the same samples and plotted dorsal versus ventral features. We found induction of the dorsal choroid plexus marker *TTR* at the expense of the ventral marker *ASCL1* (Achaete-Scute Family bHLH Transcription Factor 1) in Rm organoids (Fig 4B), indicating a deviation in dorso-ventral patterning processes resulting in the production of more dorsal tissues at the expense of more ventral tissues. To assess coregulation of genes and correlate deviation in patterning processes, we computed a correlation matrix based on the logarithmic fold change values of given genes in Rm versus Ctrl samples at d30 of organoid development (Fig 4C). These data do not only show coregulation nodes of choroid plexus markers (*TTR*, *OTX2*, *FOXA2*), ventral markers (*LMX1A*, *DLX2*, *ASCL1*), and the ventral signaling pathway SHH (*GLI1*, *PTCH1*) but also underscore that the emergence of the dorsal most structures, the choroid plexus, occurs at the expense of more ventral features. Next, we temporally dissected the development of the rostro-caudal axis by following the expression of the caudally expressed patterning gene *GBX2* by qRT-PCR. This analysis confirmed earlier bulk RNA-seq data (Fig 3A) and revealed a continuously stronger increase in the expression of *GBX2* in Rm organoids compared with Ctrls, indicating caudalization of the resulting neural tissue (Fig 4D). As a consequence of the acquisition of a more caudal regional identity and in accordance with the patterning data described, *ATOH1*, a gene highly expressed in hindbrain neurons, is strongly induced in Rm organoids at d30 (Fig 4E). In contrast, the full knock-out of *MID1* (KO) does neither show an induction of *GBX2* over time nor the increased expression of *ATOH1* at d30, again indicating different mechanisms at work.

Together, these data reveal that on a transcriptome level, the absence of the full-length isoform of MID1 when preserving the shorter isoforms results in an early patterning phenotype that subsequently causes increased dorsal and caudal regional identity, resembling the choroid plexus, possibly of the fourth ventricle. To elucidate the tissue morphology and the cellular identity induced by the Rm mutations, we stained d30 brain organoids with an antibody recognizing the choroid plexus marker TTR (Aleshire et al, 1983) (Fig 4F). We quantified and compared the area of organoid slices covered by TTR-positive structures. Whereas there was a strong up-regulation of TTR-positive areas in the Rm organoids across several batches, this was not seen in Ctrl and *MID1* KO organoids (Fig 4G). As described earlier (Lancaster et al, 2013; Pellegrini et al, 2020), TTR-positive areas, indicative of the formation of choroid plexus-like structures, are generally visible to some extent in unguided brain organoids. In line with this observation, we did detect few TTR-positive areas also in the d30 organoids generated from the Ctrl hiPSC lines, but these were much less frequent and less extensive. In vivo, the choroid plexus is a monolayered epithelial structure characterized by multiciliated cells able to

---

number of genes upregulated during differentiation from hiPSC to d5/d8/d11 in each condition and comparing the different conditions. **(G)** The Euler diagram showing the number of downregulated genes across experimental conditions during early neural differentiation. **(F, H)** Bar graph showing the top 20 GO terms significantly enriched in the genes upregulated specifically in the Ctrl organoids (i.e., 162 genes from (F)). **(F, I)** Top 20 GO terms enriched upon analysis of the Rm-specific upregulated genes, i.e., 97 genes from (F).

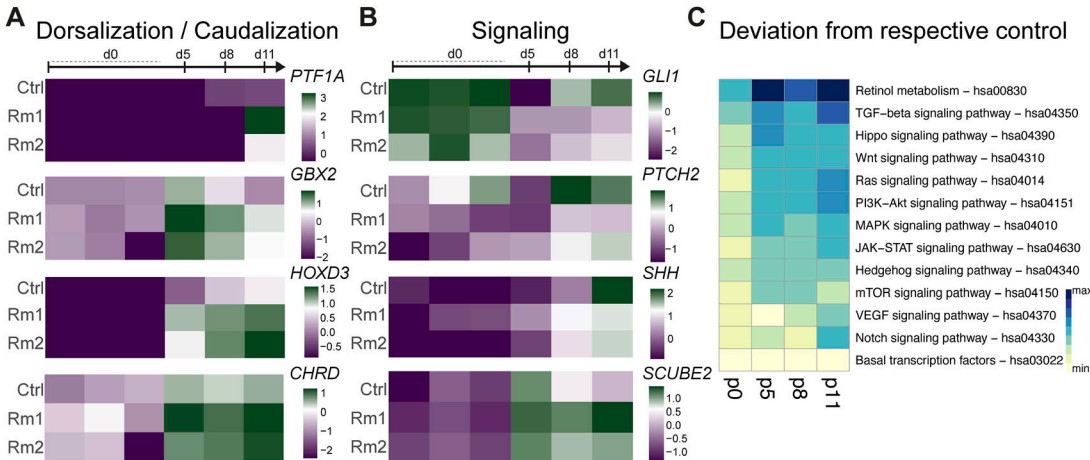

**Figure 3. Molecular alterations induced in the Rm lines upon neural induction.**
**(A)** Heatmaps depicting the scaled normalized counts per million values of different markers showing increased dorsalization and caudalization in the Rm lines across conditions and time points. **(B)** Altered expression of selected members of the sonic hedgehog pathway across conditions and timepoints, as summarized in a heatmap of the scaled normalized counts per million values. **(C)** Shown is the overall transcriptional deviation of members of KEGG signaling pathways (hsa numbers indicated) in the Rm lines compared with Ctrl at different timepoints (human induced pluripotent stem cell stage = d0, different timepoints during organoid generation = d5, 8, 11 upon aggregation of human induced pluripotent stem cells) as assessed by bulk RNA-sequencing. Note that retinol metabolism and TGF-β signaling pathways are deregulated earlier more profoundly than other signaling pathways.

produce and secret the cerebrospinal fluid. The TTR-positive structures in the Rm organoids resemble the in vivo correlate in its morphological appearance. Staining of the cilia marker ADP Ribosylation Factor Like GTPase 13B (ARL13B) confirmed that the epithelial cells within the choroid plexus-like structures in the brain organoids are largely multiciliated cells (Fig 4H). Furthermore, we found high expression of SOX9 (SRY Box Transcription Factor 9) (Fig 4I) shown to be enriched in the choroid plexus (Vong et al, 2021). The appearance of cystic areas, which may or may not originate from choroid plexus-like structures, is sometimes considered a sign of bad quality of the organoids. The extent of the formation of such cysts varies between hiPSC lines with different genetic background. To avoid such confounding variables from different genetic background in this article, all lines have the same genetic background, yet the increased appearance of choroid plexus-like structures is selectively enriched in the Rm1 and Rm2 lines, but only minor in the isogenic Ctrl and the KO lines.

In sum, our data show that Rm mutations introduced in coding exon 1 of the *MID1* gene cause a loss of the full-length isoform when maintaining the expression of shorter isoforms without a RING domain. These changes in the isoform composition of *MID1* resulted in a patterning phenotype in brain organoids, contrasting the phenotype of a full knock-out of *MID1*. Notably, the Rm organoids exhibit a striking hyper-dorsalization at the expense of ventral structures. It would be interesting to test whether early extrinsic patterning cues, for example, through addition of small molecules during brain organoid formation, would be able to counteract the intrinsic patterning defects and rescue the phenotype. The aberrant developmental patterning that we observed is highly unlikely to be compatible with proper embryonic development, thus providing a plausible explanation for the absence of patients displaying pathogenic variants in the most N-terminal region of the *MID1* gene.

# Materials and Methods

### Human fibroblasts

Fibroblasts to generate the Ctrl, Rm1, Rm2, and KO hiPSC lines were acquired at the University Medical Center in Mainz after approval by the Local Ethical Committee (No. 4485). Consent for further analysis and usage for research in an anonymized way was given. For derivation of primary fibroblasts, skin punch biopsies (4 mm) were taken in the hospital of the University Medical Center in Mainz as previously described, applying small modifications (Vangipuram et al, 2013). Briefly, biopsies were cut into small pieces containing all skin layers and plated on a six-well plate coated with 0.1% gelatin (Sigma-Aldrich) with fibroblast extraction media DMEM (Thermo Fisher Scientific), 20% FBS (Thermo Fisher Scientific), 1% penicillin/streptomycin (P/S) (Thermo Fisher Scientific). Medium was changed every other day until fibroblasts started to migrate out of the skin biopsies after 7–10 d. Cells were then transferred to two T75 flasks after 3–4 wk using TrypLE Express enzyme (Thermo Fisher Scientific). Fibroblasts were cultured in IMDM (Thermo Fisher Scientific), 15% FBS, 1% P/S. When reaching 90% confluency, fibroblasts were replated into T175 flasks and expanded as needed or frozen in liquid nitrogen for long-term storage.

### Fibroblast reprogramming into pluripotent stem cells

Male control fibroblasts were reprogrammed using commercially available Sendai viruses encoding for the four reprogramming factors OCT4, KLF4, SOX2, and CMYC ("CytoTuneTM-iPS 2.0 Sendai Reprogramming Kit"). We followed the manufacturer's instructions to use the feeder-dependent approach with small modifications. Fibroblasts were seeded in a six-well plate (10–35 × 10⁴ cells/well)

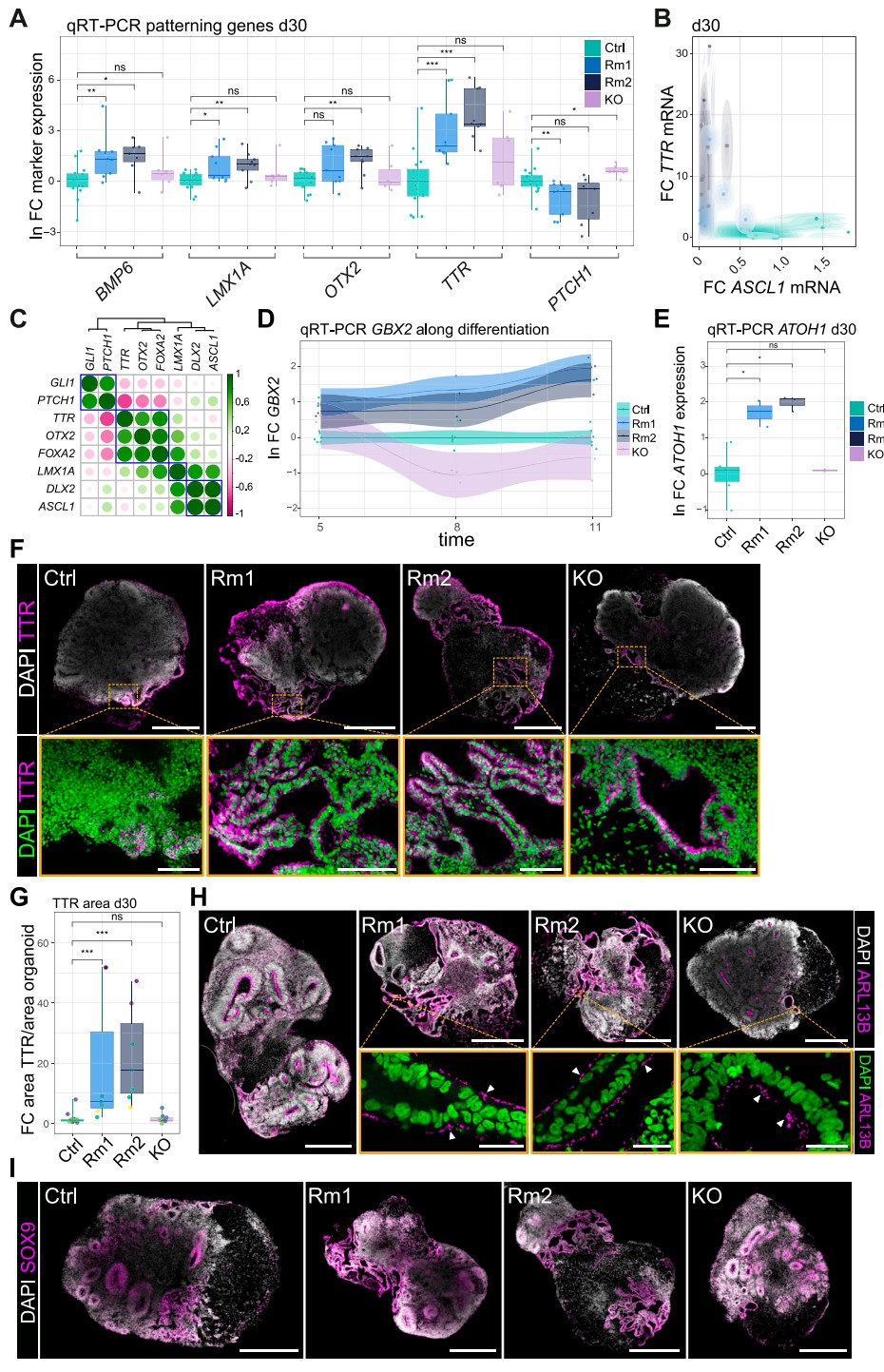

**Figure 4. MID1 N-terminal mutations Rm1 and Rm2 result in a severe patterning phenotype.**
**(A)** Box and jitter plots show the fold change of the expression of patterning genes *BMP6*, *LMX1A*, *OTX2*, *TTR*, and *PTCH1* normalized to *GAPDH* in d30 Ctrl, Rm1, Rm2, and MID1 KO organoids. Mann–Whitney-*U* test, *P < 0.05, **P < 0.01, ***P < 0.001. ns, not significant. Exact *P*-values (left to right) 0.0083, 0.014, 0.34, 0.037, 0.0096, 0.49, 0.084, 0.0014, 0.8, 0.0003, 0.00004, 0.29, 0.0053, 0.076, 0.017. (*BMP6*: Ctrl: n = 15, six batches; Rm1: n = 9, three batches; Rm2: n = 7, three batches; KO: n = 6, three batches. *LMX1A*, *OTX2*, *TTR*: Ctrl: n = 16, six batches; Rm1: n = 9, three batches; Rm2: n = 9, three batches; KO: n = 6, three batches; *PTCH1*: Ctrl: n = 16, six batches; Rm1: n = 9, three batches; Rm2: n = 7, three batches, KO: n = 6, three batches). **(B)** Brain organoids (d30) were used to quantify mRNA expression levels of *TTR* and *ASCL1* across conditions. Note the increase in *TTR* expression at the expense of the ventral marker *ASCL1* in the MID1 Rm mutant organoids. (Ctrl: n = 9, three batches; Rm1: n = 7, three batches; Rm2: n = 9, three batches). **(C)** Correlation matrix of the logarithmic fold change values of MID1 mutants versus Ctrl shows coregulated nodes and overall anti-correlation of dorsal choroid plexus marker *TTR* with the patterning genes *PTCH1* and *GLI1*. (*BMP6*: Ctrl: n = 15, six batches; Rm1: n = 9, three batches; Rm2: n = 7, three batches; *LMX1A*, *OTX2*, *TTR*, *DLX2*: Ctrl: n = 16, six batches; Rm1: n = 9, three batches; Rm2: n = 9, three batches; *PTCH1*: Ctrl: n = 16, six batches; Rm1: n = 9, three batches; Rm2: n = 7, three batches; *GLI1*: Ctrl: n = 16, six batches; Rm1; n = 9, three batches; Rm2; n = 8, three batches; *FOXA2*: Ctrl: n = 9, three batches; Rm1; n = 9, three batches; Rm2; n = 9, three batches; *ASCL1*: Ctrl: n = 9, three batches; Rm1; n = 9, three batches; Rm2; n = 8, three batches). **(D)** Line-plot showing the temporal pattern of *GBX2* misexpression (ln of FC) along early differentiation in different conditions; solid line: mean; shade: 95% confidence interval. Note the constant increase in *GBX2* in the Rm lines, contrasting the expression in the KO line (Ctrl: n = 6 from two batches, Rm1, Rm2, KO: n = 3 from one batch). **(E)** Box and jitter plots show the natural logarithm of the fold change values versus mean of Ctrl of *ATOH1* expression normalized to *GAPDH* in d30 organoids across conditions. (Ctrl: n = 6 from two batches, Rm1, Rm2, KO: n = 3 from one batch). Mann–Whitney-*U* test, *P < 0.05, ns, not significant. Exact *P*-values (top to down) 1, 0.024, 0.024. **(F)** Images show sections of d30 organoids of all conditions (Ctrl, Rm1, Rm2, KO) stained for TTR. DAPI was used to counterstain nuclei and visualize cellular organization. Orange dashed line indicates zoom-in areas shown in the lower panel. Note that the tissue positive for TTR is organized as monolayered epithelium. **(G)** Quantification of TTR-positive areas covering the area of organoid indicated as fold change over Ctrl organoids across conditions. Different batches are indicated by distinct colors. Ctrl: n = 15 from six batches, Rm1, Rm2: n = 7 from four batches, KO: n = 9 from three batches. Mann–Whitney-*U* test. ***P < 0.001; ns, not significant. Exact *P*-values (top to down) 0.77, 0.00032, 0.00092. For (A, B, D, E, G), dots represent individual organoids. Boxplots show median, quartiles (box), and range (whiskers). **(H)** Micrographs show immunofluorescence stainings using an antibody against the cilia marker ARL13B in d30 organoids. Orange dashed box highlights insets magnified in the lower panel. Arrow heads point towards multiciliated cells. **(I)** SOX9 protein expression in Ctrl, Rm1, Rm2, and MID1 KO d30 organoids. Note the high expression in ventricular zone-like structures and the choroid plexus-like areas. For (F, H, I), DAPI was used to counterstain nuclei. **(F, H, I)** Scale bars = 500 µm (F, H upper, I), 100 µm ((F), lower), 25 µm ((H), lower).

replacing the standard fibroblast medium by reprogramming fibroblast medium (RFM: DMEM, 10% ESC-qualified FBS, 1% NEAA, 0.1% β-mercaptoethanol). On day 0, Sendai virus transduction was performed on cells with 30–60% confluency. 24 h after transduction, the media was changed completely and replaced by fresh RFM. Cells were then fed every other day for the next 6 d. On day 7, after transduction, fibroblasts were transferred onto the MEF feeder cells. On day 8, medium was changed to hiPSC medium (DMEM/F-12, 20% KOSR, 1% NEAA, 0.1% β-mercaptoethanol, 1% Pen/Strep, 0.04% bFGF) and from then replaced every day, increasing the volume over time to compensate for cell growth. About 21–28 d after transduction, hiPSC colonies reached an appropriate size and were clearly visible. ~50 colonies were picked by scraping and sucking with a 200-μl pipette under a microscope placed inside the working bench. Each colony was transferred to a single well of a Matrigel-coated 12-well plate with mTeSR1 (StemCell Technologies).

## Genome editing

For CRISPR/Cas9 genome editing, hiPSCs were electroporated using the Lonza 4D-NucleofectorTM X Unit. 800,000 single cells were pelleted per transfection reaction. Cells were resuspended in 100 μl electroporation buffer P3 (P3 Primary Cell Solution Box; Lonza) plus 2.5 μg of the CAG-Cas9-Venus plasmid (pU6-(BbsI)sgRNA_CAG-Cas9-venus-bpA was a gift from Ralf Kuehn [plasmid # 86986; Addgene]) and 2.5 μg of gRNA-containing plasmid. The CAG-Cas9-Venus plasmid did not contain any gRNAs, but the gRNAs were provided by using the gRNA cloning vector (gRNA_Cloning Vector was a gift from George Church [plasmid # 41824; Addgene]). The gRNA sequences are listed in Table 1. Cell-plasmid solution was transferred to the electroporation cuvette (Amaxa P3 primary cell 4D-Nucleofector X, Lonza) and electroporated using the program CB-150 of the nucleofector. After electroporation, 100 μl of RPMI (Thermo Fisher Scientific) were pipetted into the cuvette and incubated for 10 min at 37°C and then transferred to a fresh well of a Matrigel-coated six-well plate with 2 ml of mTeSR1 (StemCell Technologies) supplemented with Y-27623 ROCK-inhibitor (10 μM, StemCell Technologies). Single GFP-positive cells were sorted in a 96-well plate with mTeSR Plus CloneR (StemCell Technologies). 14 d after plating, colonies were transferred to a 12-well plate with mTeSR Plus CloneR. A small volume (~10 μl) of the dissociated cells

was used for DNA isolation by using Quick-DNA Microprep Kit (Zymo Research). The DNA was amplified by PCR and sequenced via Sanger sequencing. Clones carrying the desired mutation were selected for further expansion.

## Generation and overexpression of GFP-tagged MID1 constructs

Overexpression constructs to analyze localization of WT and truncated MID1 proteins in HeLa cells were generated by using specific primers targeting the distinct ATGs present in the MID1 sequence. PCR products were cloned into the eGFP-C1 plasmid (Clontech). The eGFP-C1-MID1-del4 plasmid was constructed as described previously (Schweiger et al, 1999). Calcium phosphate was used to overexpress the MID1 expression constructs in HeLa cells. For this purpose, 10,000 cells were seeded per well of a 12-well plate containing a glass coverslip. 24 h after seeding, 3 μg of plasmid DNA were combined with 10 μl of CaCl2 (2.5 M) and filled up with water to a total volume of 80 μl. When being vortexed, 80 μl of 2x HEPES-Buffer (5.95$g$ HEPES, 8.18$g$ NaCl, 750 μl 1 M $Na_2HPO_4$, and 500 ml water) was added to the solution. After incubating for 30–45 min, the transfection solution was mixed by pipetting and completely used for transfecting a single well of the 12-well plate. Medium was completely changed 24 h after transfection, and cells were fixed using 4% PFA 48 h after transfection. Coverslips were transferred to slides prepared with 10 μl of mounting medium (0.5% DAPI; Vectashield). Pictures were taken using the "Echo Revolve" microscope.

## Brain organoid formation

mTeSR Plus medium (StemCell Technologies) was used to culture all hiPSC lines used in this study. Cells were grown on Matrigel-coated dishes in 5% $CO_2$ at 37°C until a confluency of 80–90% was reached. Brain organoid formation was used according to a published protocol including small adaptations (Lancaster et al, 2013). Briefly, accutase (Thermo Fisher Scientific) was used to generate single-cell suspensions of hiPSCs. After centrifugation, cells were resuspended in organoid formation medium supplied with 4 ng/ml of low bFGF (Peprotech) and 5 μM ROCK-inhibitor Y-27632 (StemCell Technologies). Organoid formation medium consisted of DMEM/F12 + GlutaMAX-I (Thermo Fisher Scientific), 20% KOSR (Thermo Fisher Scientific), 3% FBS (Thermo Fisher Scientific), and 0.1 mM

**Table 1. gRNAs used in this study.**

| gRNA sequences for hiPSCs | | Sequence |
|---|---|---|
| MID1 Exon1 gRNAs | MID1_c.Ex1_sgRNA#1_cut_r | 5'-CACCGTGAGCCCGTCTAGACCTCGC-3' |
| | MID1_c.Ex1_sgRNA#2_cut_r | 5'-CACCGATGGACTCCACAGACTCGT-3' |
| MID1 KO gRNAs | MID1_anfang_gRNA2_f | 5'-TTTCTTGGCTTTATATATCTTG TGGAAAGGACGAAACACCGAAAGCGCCCCTAATCCTCG-3' |
| | MID1_anfang_gRNA2_r | 5'-GACTAGCCTTATTTTAACTTGCTATTTCTAGCTCTAAAACCGAGGA TTAGGGGCGCTTTC-3' |
| | MID1_ende_gRNA2_f | 5'-TTTCTTGGCTTTATATATCTTGTGGAAAGGACGAAACACCGTCTA GGAGAATCCTAGCAGT-3' |
| | MID1_ende_gRNA2_r | 5'-GACTAGCCTTATTTTAACTTGCTATTTCTAGCTCTAAAACACTG CTAGGATTCTCCTAGAC-3' |

MEM-NEAA (Thermo Fisher Scientific), 0.1 mM 2-mercaptoethanol (Sigma-Aldrich). 9,000 cells in 150 μl organoid formation medium/well were aggregated in low attachment 96-well plates (Corning) for at least 48 h during which embryoid bodies (EBs) formed. After 72 h, half of the medium was replaced with 150 μl of new organoid formation medium without bFGF and ROCK inhibitor. At day 5, neural induction medium consisting of DMEM/F12 + GlutaMAX-I (Gibco), 1% N2 supplement (Gibco), 0.1 mM MEM-NEAA (Gibco), and 1 μg/ml Heparin (Sigma-Aldrich) was added to the EBs in the 96-well plate to promote their growth and neural differentiation. Neural induction medium was changed every 2 d until day 12/13, when aggregates were transferred to undiluted Matrigel (Corning) droplets. The embedded organoids were transferred to a petri dish (Greiner Bio-One) containing organoid differentiation medium without vitamin A. 3 or 4 d later the medium was exchanged with organoid differentiation medium with vitamin A, and the plates were transferred to an orbital shaker (IKA Rocker 3D digital) set to 30 rpm inside the incubator. Medium was changed twice per week. Organoid differentiation medium consisted of a 1:1 mix of DMEM/F12 + GlutaMAX-I (Thermo Fisher Scientific) and Neurobasal medium (Thermo Fisher Scientific), 0.5% N2 supplement (Thermo Fisher Scientific), 0.1 mM MEM-NEAA (Thermo Fisher Scientific), 100 U/ml penicillin and 100 μg/ml streptomycin (Thermo Fisher Scientific), 1% B27 +/− vitamin A supplement (Thermo Fisher Scientific), 0.025% insulin (Sigma-Aldrich), and 0.035% 2-mercaptoethanol (Sigma-Aldrich). For fixation, organoids were transferred from petri dishes to 1.5-ml tubes. Organoids were washed with PBS and then fixed with 1xPBS buffered 4% PFA (Carl Roth) for 30 min. Time of PFA fixation was extended up to 1 h, depending on the size of the organoids. Afterwards, organoids were washed three times for 10 min with PBS and incubated in 30% sucrose (Sigma-Aldrich) in PBS for cryoprotection. For cryosectioning, organoids were embedded in Neg-50 Frozen Section Medium (Thermo Fisher Scientific) on dry ice. Frozen organoids were cryosectioned in 30 μm sections using the Thermo Fisher Scientific Cryostar NX70 cryostat. Sections were placed on SuperFrost Plus Object Slides (Thermo Fisher Scientific) and stored at −20°C until use.

To control for batch-to-batch variation, we started the generation of organoids of different conditions (Ctrl, Rm1, Rm2, KO) together with their controls and considered them as one batch. The analysis of a given phenotype is always batch-controlled, for example, normalized to the mean value of the controls in the respective batch, thereby minimizing the impact of batch-to-batch heterogeneity and focusing on the consequences of the mutations.

## Immunocytochemistry

Citric acid antigen retrieval was performed as needed for TTR staining. Organoid slices were boiled for 5 min in 0.01 M citric acid adjusted to pH6. After replacing half of the solution with water, slices were allowed to cool down to RT for 30 min and washed once with PBS. Post-fixation of organoid slices was achieved using 4% PFA for 15 min, followed by three washing steps with PBS for 5 min. During the entire staining procedure, slides were kept in humidified staining chambers in the dark. Slices were washed briefly with blocking solution (PBS, 4% normal donkey serum [Sigma-Aldrich], 0.25% Triton-X 100 [Sigma-Aldrich]), followed by 1 h incubation with

blocking solution at RT. Primary antibodies were diluted in antibody solution (PBS, 4% normal donkey serum, 0.1% Triton-X 100), and tissue sections were incubated overnight at 4°C. Next, following three washes using PBS with 0.5% Triton-X 100, secondary antibodies were added, diluted in the antibody solution and incubated for 1 h at RT. Sections were washed three times with PBS containing 0.5% Triton-X 100 for 5 min. Slides were counterstained with DAPI 1:1,000 in PBS for 5 min, followed by one washing step with PBS. Lastly, organoid sections were mounted using Aqua Polymount (Polysciences).

Antibodies used were selected according to the antibody validation reported by the distributing companies. Rabbit anti-active Caspase-3 (G7481, 1:250; Promega), rabbit anti-ARL13B (17711-1-AP, 1:250; Proteintech), mouse (IgG1) anti-MAP2 (M4403; 1:300; Sigma-Aldrich), sheep anti-Prealbumin (AHP1837, 1:100; = TTR; Bio-Rad), rabbit anti-SOX2 (ab137385; 1:300; Abcam), rabbit anti-SOX9 (Stolt et al, 2003), 1:2,000. The following secondary antibodies were used (1:500 dilution): goat anti-rabbit Alexa 488 (A11008; Thermo Fisher Scientific), donkey anti-sheep Alexa 488 (A11015; Thermo Fisher Scientific), goat anti-mouse IgG1 Alexa 555 (A21127; Thermo Fisher Scientific), goat anti-rabbit Alexa 633 (A21070; Thermo Fisher Scientific).

## Microscopy and image analysis

Brightfield pictures were taken using the EVOS XL core on d3 of the brain organoid formation protocol. Diameters were measured using FIJI (v1.52–1.53), and the resulting values were normalized to the average value of the respective control organoids (C1 and C21) in each batch using R (v3.5.1–4.1.2).

Epifluorescence pictures were taken using the EVOS M7000 Imaging System (Thermo Fisher Scientific) and the Revolve microscope (Echo). Z-stacks were taken using an Apotome.2 (Zeiss) equipped with the Colibri 5 light source (Zeiss). Images were analyzed using FIJI (v1.52–1.53). To quantify the VZLS area, the total organoid area (excluding areas covered by cysts) and the area covered by VZLS were measured for each organoid section using FIJI (v1.52–1.53). In R (v3.5.1–4.1.2), the fraction of organoid area covered by VZLS area was calculated; these values were then averaged from different sections from the same organoids, and the resulting values were normalized to the average value of the respective control organoids (C1 and C21) in each batch. To quantify the choroid plexus-like area in organoid sections, TTR and characteristic structural features were used as markers. Therefore, the total organoid area (excluding areas covered by cysts) and the area covered by choroid plexus-like structures were measured using FIJI (v1.52–1.53). The fraction of organoid area covered by choroid plexus-like area was calculated in R (v3.5.1–4.1.2), and the resulting values were normalized to the average value of the respective control organoids (C1 and C21) in each batch. To quantify the neural area covered by SOX2 and MAP2 in organoid sections, we used OpenCV (v.4.4.0–4.5.1) in Python (v3.9.1–3.9.10) for automated thresholding (same threshold for all pictures) and counting of thresholded pixels. The neural area was determined as the number of pixels thresholded for either SOX2 or MAP2, and the fraction of SOX2/neural area and the fraction of MAP2/neural area were calculated using NumPy (v.1.21.5) and Pandas (v1.3.4).

**Table 2.  Primers used in this study.**

| qRT-PCR primer | Sequence |
|---|---|
| ASCL1_fw | 5′-AGGTGGAGACACTGCGCT-3′ |
| ASCL1_rv | 5′-CGATCACCCTGCTTCCAAAGT-3′ |
| BMP6_fw | 5′-TCAACCGCAAGAGCCTTCT-3′ |
| BMP6_rv | 5′-TCACCCTCAGGAATCTGGGA-3′ |
| DLX2_fw | 5′-GCCTCAACAACGTCCCTTACT-3′ |
| DLX2_rv | 5′-TCACTATCCGAATTTCAGGCTCA-3′ |
| GAPDH_fw | 5′-AGCCACATCGCTCAGACAC-3′ |
| GAPDH_rv | 5′-GCCCAATACGACCAAATCC-3′ |
| GLI1_fw | 5′-CAGGCTGGACCAGCTACATCA-3′ |
| GLI1_rv | 5′-TGGTACCGGTGTGGGACAA-3′ |
| LMX1A_fw | 5′-TCAGAAGGGTGATGAGTTTGTCC-3′ |
| LMX1A_rv | 5′-GGGGCGCTTATGGTCCTTG-3′ |
| MID1_Ex1-2_fw | 5′-TGTGTGACCGATGACCAGTT-3′ |
| MID1_Ex1-2_rv | 5′-GTTTTGCTTCAATTTGTCATAGC-3′ |
| MID1_Ex6-7_fw | 5′-ACCATATTCACCGGACAAGC-3′ |
| MID1_Ex6-7_rv | 5′-GGTTCTGCTTGATGTTGGGTA-3′ |
| MID1_Ex8-9_fw | 5′-CTCACACACCTGAACGCTTC-3′ |
| MID1_Ex8-9_rv | 5′-CAGACACTTGTTCCACACGG-3′ |
| OTX2_fw | 5′-GTCGAGGGTGCAGGTATGG-3′ |
| OTX2_rv | 5′-CATGCAGGAAGAGGAGGTGG-3′ |
| PTCH1_fw | 5′-CTCGCCTATGCCTGTCTAACC-3′ |
| PTCH1_rv | 5′-GATCAATGAGCACAGGCCCA-3′ |
| TTR_fw | 5′-CGGTGAATCCAAGTGTCCTCT-3′ |
| TTR_rv | 5′-GATGCCAAGTGCCTTCCAGTA-3′ |

## Quantitative RT–PCR

Total RNA was extracted using the RNeasy Mini Kit (QIAGEN), and samples were stored at –80°C until use. The RNA concentration was measured using a NanoDrop (PeqLab) or Qubit4 (Thermo Fisher Scientific). cDNA was generated starting from 125 ng up to 500 ng of total RNA with the Maxima First Strand cDNA Synthesis Kit (Thermo Fisher Scientific) or the PrimeScript RT Master Mix (Takara). In each individual experiment, equal amounts of RNA were used for the generation of cDNA. The primers used are listed in Table 2. For quantitative RT–PCR (qRT-PCR) analysis, all samples were run in triplicates each with a reaction volume of 10 μl, using the QuantiFast SYBR Green PCR Kit (Thermo Fisher Scientific) or TB Green Premix Ex Taq II (Tli RnaseH Plus) Kit (Takara), 1 μM primers, and 1 μl of cDNA. The reaction was performed in a QuantStudio 6 Flex Real-Time PCR System (Thermo Fisher Scientific) or the StepOne Plus Real-Time PCR System (Thermo Fisher Scientific) using the following amplification parameters: 5 min at 95°C, 40 cycles of 10 s at 95°C, and 1 min at 60°C. Data were analyzed using the $2^{-\Delta\Delta CT}$ method as previously described (Livak & Schmittgen, 2001), and the natural logarithm therefore was used as indicated; expression levels were obtained, normalizing each sample to the endogenous *GAPDH* control.

## Western blot

Protein lysates were generated from cell pellets using Magic Mix (48% urea, 15 mM Tris pH 7.5, 8.7% Glycerin, 1% SDS, 143 mM β-mercaptoethanol) containing protease and phosphatase inhibitors (cOmplete Tablets easypack, PhosSTOP easypack, Roche) and transferred to a QIAshredder column (QIAGEN). After centrifugation at 12,000*g* for 2 min, the solution was transferred into a fresh tube and frozen at –80°C until needed. The concentration was not measured. SDS gel electrophoresis was used to separate proteins by their size. Proteins were transferred to a PVDF membrane by using the Trans Blot Turbo Transfer Pack (Bio-Rad). Membranes were incubated for 1 h with blocking buffer (PBS, 0.1% Tween, 5% milk), followed by overnight incubation with primary antibody diluted in blocking buffer. Membranes were washed three times for 10 min with PBS-T (PBS, 0.1% Tween), incubated for 1 h with secondary antibody diluted in blocking buffer, and washed three times for 10 min with PBS-T (PBS, 0.1% Tween). Membranes were exposed using the Western Lightning Plus-ECL (Perkin Elmer), and imaging was performed by using ChemiDoc Imaging System (Bio-Rad). Images were prepared and analyzed using the Image Lab software (Bio-Rad). Mouse monoclonal anti-β-ACTIN (A2066-200UL; 1:2,000; Sigma-Aldrich), rabbit polyclonal anti-MID1 C-terminal (NBP1-26612; 1:500; Novus).

## Bulk RNA-seq

Samples were collected along the timeline of brain organoid formation. Specifically, for each hiPSC line, three wells of a six-well plate were detached using accutase. Embryoid bodies were collected before neural induction on day 5 and subsequently on days 8 and 11 of brain organoid formation. For each timepoint and hiPSC line, eight embryoid bodies were pooled. Samples were washed twice with PBS and total RNA was extracted using the RNeasy Mini Kit (QIAGEN). Poly-A enrichment-based library preparation and transcriptome sequencing were performed by Novogene Europe (GBLibraries). Paired-end 150-bp reads were sequenced on a NovaSeq6000.

The FASTQ files were preprocessed using fastp (v0.23.2) (Chen et al, 2018) with the following settings: qualified_quality_phred 20, unqualified_percent_limit 10, n_base_limit 2, length_required 20, low_complexity_filter enable, complexity_threshold 20, dedup enable, dup_calc_accuracy 6, overrepresentation_analysis, detect_adapter_for_pe, cut_right. The resulting FASTQ files were aligned to the human genome GRCH38 release 42 from GENCODE using R (v4.2.2) and the R package Rsubread (v2.12.0) (Liao et al, 2019). To count the aligned reads, we used the Rsubread built-in function *feature_count* with the following settings: (isPairedEnd=T, countReadPairs=T, requireBothEndsMapped=T, countChimericFragments=T, countMultiMappingReads=F, allowMultiOverlap=F). We then used EdgeR (Robinson et al, 2010) (v3.40.2) to filter genes using the EdgeR built-in function *filterByExpr* with default values and normalized differences in sequencing depth between samples using *calcNormFactors*. PCA analysis was performed with EdgeR and plotted using ggplot2 (v3.4.1). For the confidence ellipses we took advantage of the R package ggpubr (v.0.6.0). Differential gene expression was calculated using EdgeRs *glmQLFIT* and *glmTreat* functions.

Genes with a *P*-value < 0.01 a FDR < 0.01 and a log$_2$FC > 2 were considered as differentially expressed in the respective comparisons. Euler diagrams were plotted using the R package eulerr (v.7.0.0). GO analysis was performed using topGO (v.2.52.0) using the classic algorithm with Kolmogorov-Smirnov testing for significance and considering GO terms with a *P*-value < 0.05 and having at least five differentially expressed genes per GO term. The top 20 enriched GO terms are shown. Heatmaps were plotted using the normalized, log$_2$ transformed CPM values or the scaled normalized and log$_2$ transformed CPM values as indicated using ggplot2 (v.3.4.1). To calculate the transcriptional deviation of signaling pathways in Rm mutants the Bioconductor package KEGGREST (v.1.42) was used to retrieve the genes annotated in a given KEGG pathway. Genes not present in our sequencing data were not considered. For each gene and differentiation timepoint the mean of Rm samples was divided by the mean of the controls to calculate the fold change. This fold change was next log transformed (natural logarithm) and the absolute value calculated, so that deviation in any direction has the same impact on the calculated score. These scores of all individual gene were then summed up and divided by the number of genes to normalize for different number of annotated genes in individual pathways. Basic R (v.4.3.2) functions were used for the calculations and the R package pheatmap (v 1.0.12) was used for plotting.

### In silico pathogenicity score assessment

To assess the predicted effect of genetic variants in the *MID1* gene, REVEL scores for all biologically possible non-synonymous single nucleotide variants of the canonical *MID1* transcript NM_000381.4 were annotated as previously demonstrated (Schroter et al, 2023). The geom_smooth function of the R ggplot2 package was used to display a smoothened line and confidence interval of all REVEL scores mapped to their individual position of a linearized version of the MID1 protein.

### Statistics and reproducibility

Data were statistically analyzed with Microsoft Excel, GraphPad Prism, or R using statistical tests indicated throughout the manuscript. No statistical methods were used to predetermine sample size. The investigators were not blinded to allocation and outcome analysis. The experiments were not randomized.

## Data Availability

The FASTQ files of bulk RNA-seq data have been deposited in the European Nucleotide Archive (ENA) at EMBL-EBI under accession number PRJEB71585. The data that support the findings of this study are available from the corresponding authors upon reasonable request.

## Supplementary Information

## Acknowledgements

We thank Michael Wegner (FAU Erlangen-Nürnberg) for sharing equipment and laboratory space. This work was supported by grants from the German Research Foundation (CRC1080, project number 221828878), the ReALity Excellence Initiative (SCHW 829/7-1) of the University Mainz to S Schweiger, German Research Foundation (KA3125/2-1; GFK2162/2 TP C2), Schram Foundation (T287/29577/2017) to M Karow, the Bavarian State Ministry of Sciences, Research, and the Arts (ForInter; F.2-F2412.30/1/24) to M Karow and S Falk, Interdisciplinary Center for Clinical Research (IZKF) at the University Hospital of the FAU Erlangen-Nürnberg to M Karow (P068; Jochen-Kalden funding programme N7) and S Falk (P074, E32).

### Author Contributions

S Frank: investigation.
E Gabassi: investigation.
S Käseberg: investigation.
M Bertin: investigation.
L Zografidou: investigation.
D Pfeiffer: investigation.
H Brennenstuhl: investigation.
S Falk: conceptualization, formal analysis, supervision, visualization, project administration, and writing—original draft, review, and editing.
M Karow: conceptualization, formal analysis, supervision, funding acquisition, visualization, project administration, and writing—original draft, review, and editing.
S Schweiger: conceptualization, supervision, funding acquisition, project administration, and writing—review and editing.

### Conflict of Interest Statement

The authors declare that they have no conflict of interest.

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
