## [Reviewer comments · Life Science Alliance]

Life Science Alliance

Absence of the RING domain in MID1 results in patterning defects in the developing human brain

Marisa Karow, Sarah Frank, Elisa Gabassi, Stephan Käseberg, Marco Bertin, Daniela Pfeiffer, Heiko Brennenstuhl, Sven Falk, and Susann Schweiger

DOI: <https://doi.org/10.26508/lsa.202302288>

Corresponding author(s): Marisa Karow, Friedrich-Alexander-University Nürnberg-Erlangen and Susann Schweiger, University Medical Center Mainz

Review Timeline:

Submission Date:	2023-07-24
Editorial Decision:	2023-09-05
Revision Received:	2023-12-04
Editorial Decision:	2023-12-21
Revision Received:	2023-12-28
Accepted:	2023-12-29

Transaction Report:

September 5, 2023

Re: Life Science Alliance manuscript #LSA-2023-02288-T

Prof. Marisa Karow
Friedrich-Alexander-University Nürnberg-Erlangen
Institute of Biochemistry
Fahrstrasse 17
Erlangen, Bavaria 91054
Germany

Dear Dr. Karow,

Thank you for submitting your manuscript entitled "Absence of the RING domain in MID1 results in severe patterning defects in the developing human brain" to Life Science Alliance. The manuscript was assessed by expert reviewers, whose comments are appended to this letter. We invite you to submit a revised manuscript addressing the Reviewer comments.

Thank you for this interesting contribution to Life Science Alliance. We are looking forward to receiving your revised manuscript.

Sincerely,

B. MANUSCRIPT ORGANIZATION AND FORMATTING:

Reviewer #1 (Comments to the Authors (Required)):

In this article, Frank and colleagues investigate the role of the RING domain bearing MIND isoform in cortical organoid generation and development. By using genome edited iPS cells, they demonstrate substantial neurogenesis defects due to patterning abnormalities that occur very early on. These defects lead to abundant formation of choroid plexus at the expense of cortical tissue. The manuscript is well presented and the data are convincing. In particular the evidence for early patterning defects and differentiation into choroid plexus structure is very clear. The precise cause of these phenotypes is less evident and some possibilities should be more openly discussed.

Major points:

The presence of choroid plexus is a recurrent phenomenon in cortical organoids generation, in particular in unguided protocols as mentioned by the authors. The abundance of choroid plexus can be variable between organoid batches and is often considered as a readout of batch quality. It is therefore hard to conclude here whether RING-containing MID1 is directly acting as a regulator of patterning, or whether this effect may be specific to the organoid system, but have no validity in vivo. The presence in 2 independent mutants displaying the same phenotype is a reassuring point. This is a challenging comment to address, but it would be at least important to discuss this point in the manuscript and openly report this possibility.

The description of the mutation throughout the text is misleading:

"when the RING domain of MID1 is mutated but not when MID1 is fully lost..." or "In sum our data show that mutations in the RING domain encoding exon 1 of the MID1 gene cause truncated MID1 protein variants..."

This is not the deletion of a domain leading to a truncated protein, but rather the KO of the full-length isoform, that leaves the other shorter isoforms unaffected. These shorter isoforms are naturally present in the wild type.

In this regard, the reason for the absence of phenotype in the full KO is very difficult to interpret. The only difference in the two conditions is the presence of the short isoforms in the Rm mutants and so it appears that it is the presence of these isoforms in the absence of the full length that causes the phenotype. This brings back to the first comment, namely that the phenotype in these lines could be due to MID1-independent effects. A rescue with the full length would be the way to address this, but these considerations should at least be properly discussed in the manuscript.

Other points:

Figure 1F: Can the authors verify that these structures are indeed microtubules (and not other types of filaments)? If these are indeed microtubules, they appear bundled. Could the authors compare MT organization with the wild-type? The legend/color code of figure 1F is unclear.

Figure 3D: Could the authors show lower mag images to validate that TTR is specific to the choroid plexus structures in their samples?

Reviewer #2 (Comments to the Authors (Required)):

In the manuscript entitled: "Absence of the RING domain in MID1 results in severe patterning defects in the developing human brain" Frank and colleagues aimed to model Opitz BBB/G syndrome by scrutinizing the functional role of the X-linked gene MID1. Interestingly they decided to focus on the N-terminal RING domain which although has a high probability of disease mutations there are not known patients with mutations in this region so far. They use crispr/cas9 genome-edited iPSCs, they generate unpatterned brain organoids and they follow the very early stages of human brain development using both immunofluorescence and transcriptomic analysis. They show that in mutant RING domain lines there are prominent neurogenic deficits with a reduction of neural tissue and a concomitant increase in choroid plexus-like structures a phenotype that is not present in the full KO of MID1 gene. They suggest that the N-terminal domain of MID1 regulates with a distinct mechanism the

early patterning steps during brain development.

This work is novel and very interesting since it can contribute to the limited knowledge we have so far on the molecular mechanisms that regulate early human brain patterning. The manuscript is very well written and the work has been performed in a thorough and well-designed way. However, I believe that the authors should include more details, discussion and more information/analysis to investigate further the original transcriptomic data they show. I have a few suggestions that I believe would be a useful improvement to the story.

Major comments:

- The expression of MID1 should be performed more thoroughly. The authors mention that MID1 is expressed in progenitors however from the picture they present it seems that there is a more general expression throughout the VZ- and CP-like structures in the organoids. However, the stage of the organoids selected is very young - not a thick CP-like region has been generated yet. How the MID1 is expressed in older organoids? Is there a clear accumulation only in the progenitors' area? The authors should include stainings from other organoid stages. Also, the authors should show the RNA levels either from their bulk RNA sequencing data or from other published datasets.
- The RNA sequencing data are very interesting. However, the complete list of the differentially regulated genes is not presented. Are there different genes that are dysregulated at different developmental time points? Which is the key developmental stage that is affected and from which genes the phenotype is driven? The authors mention a couple of patterning genes however there should be some validation of the relative expression of some of the key genes that are dysregulated. The authors show some of the ventral and choroid plexus markers but in much later time points (d30) compared to when they performed the transcriptomic analysis. However, it would be important to show the developmental trajectory of these organoids via the expression of some of the key genes/patterning factors in different stages.
- The authors suggest that there is an increase in the choroid plexus identity based both on their transcriptomics and stainings. However, it is not clear to me why they say that it "resembles the choroid plexus of the fourth ventricle". Is there any specific marker for the plexus of the fourth vs. the rest of the LVs' plexuses that made the authors focus on that particular plexus? If yes could the authors show the expression of such markers? This point should be discussed in more detail.
- Have the authors thought of performing some rescue experiments? For instance, the authors could think of using forebrain/dorsal-ventral patterning of their organoids to see if this will improve the neuroepithelial that will be generated in RM1 and RM2 mutants. Alternatively, they could think of manipulating some of the differentially regulated genes they found in the transcriptomic data.
- Another point that is of question reading this work is why mutations in the N-terminal region could result in such severe phenotype. Are there any known interactions of MID1 via the N-terminal regions? Which are they? How their expression is affected in the mutant lines? The authors should at least discuss more this issue and think of investigating this a bit further.

Minor comments:

- "In controls as well as in Rm1 and Rm2 additional polypeptides translated from ATG3-5 giving rise to 64 kDa, 58 kDa, and 57 kDa MID1 proteins respectively were detectable. ATG2 produced 69 kDa MID1 protein only detectable in Rm2 (Figs 1B, D)". In Figure 1B is written Rm1. Could you please check and correct it accordingly?
- Statistical analysis is missing in the manuscript. Please indicate, maybe in each figure legend, the statistical analysis that was performed before generating each graph. Also, the number of biological replicates should be included.

Point-by-Point Letter to the Reviewers

Reviewer 1

In this article, Frank and colleagues investigate the role of the RING domain bearing MIND isoform in cortical organoid generation and development. By using genome edited iPSC cells, they demonstrate substantial neurogenesis defects due to patterning abnormalities that occur very early on. These defects lead to abundant formation of choroid plexus at the expense of cortical tissue. The manuscript is well presented and the data are convincing. In particular the evidence for early patterning defects and differentiation into choroid plexus structure is very clear. The precise cause of these phenotypes is less evident and some possibilities should be more openly discussed.

We are grateful for the constructive comments of the reviewer and will answer to the comments one by one below.

- The presence of choroid plexus is a recurrent phenomenon in cortical organoids generation, in particular in unguided protocols as mentioned by the authors. The abundance of choroid plexus can be variable between organoid batches and is often considered as a readout of batch quality. It is therefore hard to conclude here whether RING-containing MID1 is directly acting as a regulator of patterning, or whether this effect may be specific to the organoid system, but have no validity in vivo. The presence in 2 independent mutants displaying the same phenotype is a reassuring point. This is a challenging comment to address, but it would be at least important to discuss this point in the manuscript and openly report this possibility.

We thank the reviewer for the comment. In fact, it is true that the appearance of cystic areas, which may or may not be choroid plexus-like structures is sometimes considered as bad quality of organoids. The extent of the formation of such cysts varies between hiPSC lines with different genetic background. To avoid such confounding variables from different genetic background in this manuscript, all lines have the same genetic background, yet the increased appearance of choroid plexus-like structures is selectively enriched in the Rm1 and Rm2 lines, but only minor in the isogenic ctrl and the KO lines. To further control for variation between different batches of organoids we started the generation of organoids of different conditions (ctrl, Rm1, Rm2, KO) together and considered them as one batch. Moreover, the analysis of a given phenotype is always batch-controlled, i.e. normalized to the mean value of the controls in the respective batch thereby minimizing the impact of batch-to-batch heterogeneity and focusing on the consequences of the mutations. We now added this information in more detail in the manuscript. Moreover, the phenotypes described are consistent throughout independent batches. We now indicated the batches analyzed in the quantification of TTR positive areas in Figure 4G. Another assuring point is that we see early on, even at the stage of hiPSCs that the global transcriptome of the Rm lines changes as compared to the ctrl and KO line and in particular signaling pathways involved in patterning are affected.

We now extended the discussion on this matter in the manuscript as suggested by the reviewer.

- The description of the mutation throughout the text is misleading: "when the RING domain of MID1 is mutated but not when MID1 is fully lost..." or "In sum our data show that mutations in the RING domain encoding exon 1 of the MID1 gene cause truncated MID1 protein variants..." This is not the deletion of a domain leading to a truncated protein, but rather the KO of

the full-length isoform, that leaves the other shorter isoforms unaffected. These shorter isoforms are naturally present in the wild type.

The comment of the reviewer raises an important aspect that we also tried to describe in the manuscript and we agree that some sections were misleading in the original manuscript. It is true that the Rm mutations lack the full-length isoform of MID1 and other isoforms are unaffected, with a 69 kDa isoform present only in the Rm1 mutant due to the 2-bp deletion that puts the ATG2 in-frame with the rest of the protein (Fig1B). As suggested by the reviewer we changed the respective passages in the manuscript text.

- In this regard, the reason for the absence of phenotype in the full KO is very difficult to interpret. The only difference in the two conditions is the presence of the short isoforms in the Rm mutants and so it appears that it is the presence of these isoforms in the absence of the full length that causes the phenotype. This brings back to the first comment, namely that the phenotype in these lines could be due to MID1-independent effects. A rescue with the full length would be the way to address this, but these considerations should at least be properly discussed in the manuscript.

As the reviewer rightly stated the difference between the Rm and the KO lines is the presence of the shorter isoforms in the absence of the full-length MID1. Importantly, MID1 has been shown to be able to homo- and heterodimerize with itself and MID2 (Short et al., 2002) and this feature is encoded in the more C-terminal coiled-coil domain. Thus, the phenotype may partially be a result of a relative shift in the formation of different dimers composed of the shorter isoforms lacking the full-length and this may lead to the observed phenotype. Importantly, the RING domain only present in the full-length MID1 protein harbors polyubiquitination activity while the B-Boxes present also in shorter isoforms perform monoubiquitination. It appears possible that the loss of polyubiquitination while maintaining monoubiquitination activity and microtubule association results in the described phenotypes. We added a section discussing this aspect to the manuscript.

MID1-independent effects, in our opinion, seem at least unlikely based on the strongly controlled background, i.e. performing all experiments in cells with the same isogenic background and the only variable changing between the different lines are the modifications in the *MID1* gene.

- Figure 1F: Can the authors verify that these structures are indeed microtubules (and not other types of filaments)? If these are indeed microtubules, they appear bundled. Could the authors compare MT organization with the wild-type? The legend/color code of figure 1F is unclear.

We thank the reviewer for this comment and performed co-staining with markers of the Tubulin cytoskeleton to show association with microtubules following overexpression of the MID1-GFP deletion constructs. We replaced the old images with these images in Fig. 1F.

We are sorry that the color code is misleading. It is a consistent color code throughout all figures. We introduce the colors in Fig 1B and kept them consistent throughout the manuscript. Given that the color code was not easily understandable we adapted the explanation of the color code in the legend of Figure 1.

- Figure 3D: Could the authors show lower mag images to validate that TTR is specific to the choroid plexus structures in their samples?

We agree with the reviewer and included also lower magnification images showing the TTR staining in whole organoid slices. New Fig 4F.

Reviewer 2

In the manuscript entitled: "Absence of the RING domain in MID1 results in severe patterning defects in the developing human brain" Frank and colleagues aimed to model Opitz BBB/G syndrome by scrutinizing the functional role of the X-linked gene MID1. Interestingly they decided to focus on the N-terminal RING domain which although has a high probability of disease mutations there are not known patients with mutations in this region so far. They use crispr/cas9 genome-edited iPSCs, they generate unpatterned brain organoids and they follow the very early stages of human brain development using both immunofluorescence and transcriptomic analysis. They show that in mutant RING domain lines there are prominent neurogenic deficits with a reduction of neural tissue and a concomitant increase in choroid plexus-like structures a phenotype that is not present in the full KO of MID1 gene. They suggest that the N-terminal domain of MID1 regulates with a distinct mechanism the early patterning steps during brain development.

This work is novel and very interesting since it can contribute to the limited knowledge we have so far on the molecular mechanisms that regulate early human brain patterning. The manuscript is very well written and the work has been performed in a thorough and well-designed way. However, I believe that the authors should include more details, discussion and more information/analysis to investigate further the original transcriptomic data they show. I have a few suggestions that I believe would be a useful improvement to the story.

We thank the reviewer for highlighting the importance of our data and are happy to include more details as suggested and summarized below.

- The expression of MID1 should be performed more thoroughly. The authors mention that MID1 is expressed in progenitors however from the picture they present it seems that there is a more general expression throughout the VZ- and CP-like structures in the organoids. However, the stage of the organoids selected is very young - not a thick CP-like region has been generated yet. How the MID1 is expressed in older organoids? Is there a clear accumulation only in the progenitors' area? The authors should include stainings from other organoid stages.

We are happy to elude on the expression of MID1 in more detail. Through immunofluorescent stainings, we showed that MID1 protein is detectable in cells within the VZ-like structures, but also in the neuron containing CP-like layer. It is true that at the rather young stage of the organoids (d30) the neuronal layer is less developed. To address this we now also included in the suppl Figure (new Fig S1F) an image of an older organoid generated from a control human pluripotent stem cell line. The organoid age is 2 month and at this stage the expression of MID1 is very apparent in progenitors at the apical side of the VZ like structures as well as in the neuronal layer above.

- Also, the authors should show the RNA levels either from their bulk RNA sequencing data or from other published datasets.

We have included the RNA expression levels of *MID1* and its homologue *MID2* (and other selected genes) in the hiPSC lines assessed by bulk RNA-seq in the supplementary Fig S2B. Moreover, we now included a temporally resolved expression profile of *MID1* during early steps of differentiation at d5, 8, 11 in the new Fig S2C.

In addition, we want to refer to a study from our laboratory which is deposited on a preprint server (Käseberg et al., 2023 BioRxiv doi.org/10.1101/2023.06.17.545424) and in which *MID1* expression is studied on a single cell level in human brain organoids. As shown in Figure 5D of this study, we observe 2 peaks of *MID1* expression in progenitors as well as in neurons.

[Figure removed by editorial staff per authors' request]

- The RNA sequencing data are very interesting. However, the complete list of the differentially regulated genes is not presented. Are there different genes that are dysregulated at different developmental time points? Which is the key developmental stage that is affected and from which genes the phenotype is driven? The authors mention a couple of patterning genes however there should be some validation of the relative expression of some of the key genes that are dysregulated. The authors show some of the ventral and choroid plexus markers but in much later time points (d30) compared to when they performed the transcriptomic analysis. However, it would be important to show the developmental trajectory of these organoids via the expression of some of the key genes/patterning factors in different stages.

We thank the reviewer for this constructive comment.

We now included the complete list of dysregulated genes in all conditions (see Table S1).

Concerning the patterning genes, as mentioned by the reviewer, we had included mRNA expression levels of ventral markers and signaling pathway members (*DLX2*, *GLI1*, *PTCH1*) as well as choroid plexus markers (*BMP6*, *LMX1A*, *OTX2*, *TTR*) in d30 organoids (previous Figs 3B, C). To resolve this in more detail we now generated a new Figure 3, dedicated to the molecular changes induced by the Rm mutants early upon inducing neural induction.

What becomes very apparent now is the aberrant induction of caudal (new Fig 3A, *HOXD3*) and hindbrain identity (*PTF1A* and *GBX2*) as well as dorsal identity (*CHRD*) genes. In addition, we included expression analyses of selected members of the SHH signaling pathway (*GLI1*, *PTCH2*, *SHH*, *SCUBE*), all indicating downregulation of pathway activity in the Rm lines compared to control. From this we conclude that the key developmental stage that is affected must be early before forcing neural induction in the embryoid bodies. We furthermore included a new insightful analysis (new Fig 3C) on the extent of the overall transcriptional deviation of KEGG signaling pathways and thereby found that Retinol (RA) metabolism as well as TGF-beta signaling pathway are even earlier and more profoundly changed in the Rm lines compared to ctrl pointing towards a sequence of event where ventral SHH is missing due to earlier changes in RA and TGF-beta signaling.

As rightly requested by the reviewer we now also verified the bulk RNA-seq data by qRT-PCR at different developmental timepoints and chose *GBX2* as an example. As shown in the new Fig 4D, we confirmed the increased expression of *GBX2* along neural differentiation underscoring the earlier findings of a caudalization in the Rm organoids. As a consequence of this early caudalization we see the strong induction of genes associated with hindbrain (cerebellum)

neurons (*ATOH1*) at later stages. Furthermore, we show in a correlation matrix (Fig 4C) that the pattern of expression of dorsal choroid plexus genes anticorrelates with the expression of ventral patterning genes.

- The authors suggest that there is an increase in the choroid plexus identity based both on their transcriptomics and stainings. However, it is not clear to me why they say that it "resembles the choroid plexus of the fourth ventricle". Is there any specific marker for the plexus of the fourth vs. the rest of the LVs' plexuses that made the authors focus on that particular plexus? If yes could the authors show the expression of such markers? This point should be discussed in more detail.

We agree with the reviewer that this statement was too strong in the original manuscript. The distinct induction of caudal genes such as *HOXD3* (Fig 3A), *HOXB4*, *HOXB5* (Fig S2F) but also *GBX2* and *ATOH1* prompted us to conclude the acquisition of a more caudal regional identity in Rm mutants which aligns well with the choroid plexus in the fourth ventricle. However, we agree that there is no exclusive marker for the choroid plexus in the fourth ventricle and thus down-toned this comment in the paper.

- Have the authors thought of performing some rescue experiments? For instance, the authors could think of using forebrain/dorsal-ventral patterning of their organoids to see if this will improve the neuroepithelial that will be generated in RM1 and RM2 mutants. Alternatively, they could think of manipulating some of the differentially regulated genes they found in the transcriptomic data.

We fully agree with the reviewer that these are very interesting experiments to do, however given the 3-month deadline we received for the submission of a revised version, we feel that this exceeds the scope of this study. It would require a systematic testing of the timing of the ventralization experiments (including confirmation of the effect) and dissection of the consequences by both molecular and cellular phenotyping. If we were to infer with the expression of some of the putative drivers of the phenotype, ideally we would need to use a CRISPR-based approach which would require also clonal selection of the ctrl and the disease lines. These experiments are very interesting but will need to be addressed in future studies.

- Another point that is of question reading this work is why mutations in the N-terminal region could result in such severe phenotype. Are there any known interactions of MID1 via the N-terminal regions? Which are they? How their expression is affected in the mutant lines? The authors should at least discuss more this issue and think of investigating this a bit further.

We agree with the reviewer that this is an interesting question. As also the other reviewer mentioned earlier, it must be a combinatorial effect, i.e. absence of the full-length isoform while maintaining the expression of the shorter isoforms leading to a putative gain of a different function. The N-terminus of MID1 harbors the RING finger domain entailing E3 ubiquitin ligase activity which mediates the polyubiquitination of target proteins. This activity is missing in the Rm mutants while e.g. the monoubiquitination activity enclosed in the B-Boxes is still present. The coiled-coil domain within the MID1 protein has been shown to be responsible for homo- and heterodimerization and the C-terminus for microtubule-binding. In the HeLa cell overexpression experiment (Fig 1F) we see that the N-terminally truncated proteins still bind to microtubules as in contrast to C-terminal mutations. But they lack the RING finger, thus their ubiquitin ligase activity is strongly altered. There are only few proteins (*UBE2D1*, *UBE2D2*, *UBE2D3* and *UBE2N*) reported to interact with MID1 via the RING finger domain. We did analyze their expression

dynamics but did not see any alterations. We include a hierarchical clustering plot for the reviewers which shows that the expression dynamics cluster with differentiation, i.e. age of the samples rather than with lines there are derived of and hence does not provide a basis for the explanation of the phenotypes observed.

Importantly, MID1 has been shown to be able to homo- and heterodimerize with itself and MID2 (Short et al., 2002) and this feature is encoded in the more C-terminal coiled-coil domain. Thus, the phenotype may partially be a result of a relative shift in the formation of different dimers composed of the shorter isoforms lacking the full-length and its polyubiquitination activity. It appears possible that the loss of polyubiquitination while maintaining monoubiquitination activity and microtubule association in these dimers results in the described phenotypes. We added a section discussing this aspect to the manuscript.

- "In controls as well as in Rm1 and Rm2 additional polypeptides translated from ATG3-5 giving rise to 64 kDa, 58 kDa, and 57 kDa MID1 proteins respectively were detectable. ATG2 produced 69 kDa MID1 protein only detectable in Rm2 (Figs 1B, D)". In Figure 1B is written Rm1. Could you please check and correct it accordingly?

We highly appreciate that the reviewer raised this point. In fact, this was a mistake which is now corrected in the new version of the manuscript. We apologize for this mistake.

- Statistical analysis is missing in the manuscript. Please indicate, maybe in each figure legend, the statistical analysis that was performed before generating each graph. Also, the number of biological replicates should be included.

We apologize for not including the relevant statistical information in the earlier version of the manuscript and fully agree with the reviewer that this is very important. We now included all related information in the respective figure legends.

December 21, 2023

RE: Life Science Alliance Manuscript #LSA-2023-02288-TR

Prof. Marisa Karow
Friedrich-Alexander-University Nürnberg-Erlangen
Institute of Biochemistry
Fahrstrasse 17
Erlangen, Bavaria 91054
Germany

Dear Dr. Karow,

Thank you for submitting your revised manuscript entitled "Absence of the RING domain in MID1 results in patterning defects in the developing human brain". We would be happy to publish your paper in Life Science Alliance pending final revisions necessary to meet our formatting guidelines.

- please upload your main manuscript text as an editable doc file
- please add ORCID ID for the secondary corresponding author--they should have received instructions on how to do so
- please add the Twitter handle of your host institute/organization as well as your own or/and one of the authors in our system
- please add a Summary Blurb/Alternate Abstract to our system
- please upload your main and supplementary figures as single files
- please add your main, supplementary figure, and table legends to the main manuscript text after the references section
- please remove figures from the manuscript text. All figure files should be uploaded as individual ones, including the supplementary figure files; all figure legends should only appear in the main manuscript file
- Please upload your tables in editable .doc or excel format; they can be included at the bottom of the main manuscript file or sent separately.
- please add a callout for Figure 2I to your main manuscript text
- please provide accession information for the RNA-seq data in the Data Availability statement

A. FINAL FILES:

B. MANUSCRIPT ORGANIZATION AND FORMATTING:

Sincerely,

Reviewer #1 (Comments to the Authors (Required)):

The authors have addressed most of my concerns. I therefore suggest publication of this very nice work.

Reviewer #2 (Comments to the Authors (Required)):

The revised manuscript by Sarah Frank and colleagues is substantially improved. The authors addressed most of my comments and those from the other reviewer. I do not have any other comments. I found that this manuscript is now suitable for publication.

December 29, 2023

RE: Life Science Alliance Manuscript #LSA-2023-02288-TRR

Prof. Marisa Karow
Friedrich-Alexander-University Nürnberg-Erlangen
Institute of Biochemistry
Fahrstrasse 17
Erlangen, Bavaria 91054
Germany

Dear Dr. Karow,

Thank you for submitting your Research Article entitled "Absence of the RING domain in MID1 results in patterning defects in the developing human brain". It is a pleasure to let you know that your manuscript is now accepted for publication in Life Science Alliance. Congratulations on this interesting work.

DISTRIBUTION OF MATERIALS:

Again, congratulations on a very nice paper. I hope you found the review process to be constructive and are pleased with how the manuscript was handled editorially. We look forward to future exciting submissions from your lab.

Sincerely,
